# A Systematic Review of the Association of Skipping Breakfast with Weight and Cardiometabolic Risk Factors in Children and Adolescents. What Should We Better Investigate in the Future?

**DOI:** 10.3390/nu11020387

**Published:** 2019-02-13

**Authors:** Alice Monzani, Roberta Ricotti, Marina Caputo, Arianna Solito, Francesca Archero, Simonetta Bellone, Flavia Prodam

**Affiliations:** 1SCDU of Pediatrics, Department of Health Sciences, University of Piemonte Orientale, 28100 Novara, Italy; alice.monzani@gmail.com (A.M.); roberta.ricotti@uniupo.it (R.R.); arisolito@gmail.com (A.S.); francesca.archero@gmail.com (F.A.); simonetta.bellone@med.uniupo.it (S.B.); 2Endocrinology, Department of Translational Medicine, University of Piemonte Orientale, 28100 Novara, Italy; marina.caputo@hotmail.com; 3Interdisciplinary Research Center of Autoimmune Diseases, University of Piemonte Orientale, 28100 Novara, Italy

**Keywords:** children, adolescents, breakfast, skipping, obesity, metabolic syndrome

## Abstract

The incidence of skipping breakfast in pediatric subjects is rising, and a relationship with overweight (OW) and obesity (OB) has been shown. Associations with cardiovascular outcomes and skipping breakfast in adults have been reported. The purpose of this systematic review was to summarize the association of skipping breakfast with body weight and metabolic outcomes in the pediatric population. We searched relevant databases (2008–2018) and identified 56 articles, of which 39 were suitable to be included, basing on inclusion criteria (observational; defined breakfast skipping; weight and/or metabolic outcomes). Overall, 286,804 children and adolescents living in 33 countries were included. The definitions of OW/OB, skipping breakfast, and the nutrient assessment were highly heterogeneous. Confounding factors were reported infrequently. The prevalence of skipping breakfast ranged 10–30%, with an increasing trend in adolescents, mainly in girls. Skipping breakfast was associated with OW/OB in the 94.7% of the subjects. The lack of association was shown mainly in infants. Moreover, 16,130 subjects were investigated for cardiometabolic outcomes. Skipping breakfast was associated with a worse lipid profile, blood pressure levels, insulin-resistance, and metabolic syndrome. Five studies reported a lower quality dietary intake in breakfast skippers. This review supports skipping breakfast as an easy marker of the risk of OW/OB and metabolic diseases, whether or not it is directly involved in causality. We encourage intervention studies using standardized and generalizable indicators. Data on confounders, time of fasting, chronotypes, and nutrition quality are needed to establish the best practice for using it as a tool for assessing obesity risk.

## 1. Introduction

Childhood obesity (OB) is a major public health issue in both developed and developing countries across the world [1]. Overweight (OW) and OB result from a complex network in which several factors interplay, such as genetic implications, birth weight, breastfeeding, parental obesity, physical activity, socioeconomic status, age, and gender [2]. Among them, dietary habits certainly also play a role [3]; in particular breakfast, the first meal in the day, has a critical role in energy balance and dietary regulation [4]. Despite this, the incidence of skipping breakfast among children and adolescent is rising [4,5], and several studies have shown a positive relationship between breakfast skipping and OW/OB [6]. As a possible explanation, children who regularly have breakfast have been shown to be more likely to have a better diet quality and a higher intake of key food groups, such as fruit, dairy, and dietary fibers and, furthermore, they are also more likely to meet the recommendations for micronutrients [7,8,9]. Children who skip breakfast, instead, tend to eat more energy-dense food such as fast food leading to excess hunger and overeating [10].

The evidence on the association of breakfast consumption with body weight in the European population has been collected in the systematic review published in 2010; collectively, the data from observational studies carried out in Europe until 2009 have consistently demonstrated that children and adolescents who eat breakfast have a reduced risk of becoming overweight or obese and have a lower Body Mass Index (BMI) compared with those who skip breakfast [6]. Moreover, a series of studies have reported that breakfast skipping is associated with hypertension, cardiometabolic disease, insulin insensitivity, diabetes mellitus, and mortality [11]. However, these metabolic outcomes have not been explored in a larger systematic review and this associated has not been confirmed. 

The aim of our systematic review is to analyze the association of skipping breakfast, methodologically defined based on reported questionnaires, with body weight and metabolic outcomes in the pediatric population, focusing on the studies published in the last ten years. 

## 2. Methods

### 2.1. Literature Search

PICO methodology (Population: children and adolescents; Exposure: skipping breakfast; Comparison: not skipping breakfast; Outcomes; weight and metabolic parameters) to develop a search strategy based on medical subject headings (MeSH) and keywords were used. Guidelines of the Preferred Reporting for Systematic Reviews and Meta-Analyses (PRISMA) statement were followed, and a PRISMA checklist were followed.

The Cochrane Central Register of Controlled Trials, PubMed, CINHAHL, and EMBASE databases (January 2008–December 2018) were systematically used. The reference lists of identified studies and key review articles, including previously published reviews, were also searched for all randomized and non-randomized clinical trials as well as prospective cohort and cross-sectional studies assessing the association of breakfast skipping or consumption, however, defined, with body weight and cardiometabolic aspects in children and adolescents. No country restrictions were imposed.

The search terms used included “breakfast”, and “children$” (or “adolescents$”). The search strategy used both keywords and MeSH terms. No further limitations were made so the search terms would be as sensitive as possible. In addition, we checked the references of eligible articles for further papers that were not captured by our search strategy and we corresponded with authors when the relevant information was missing in the paper.

### 2.2. Outcome Measures

The primary outcome measures were as follows: body weight, body weight excess (e.g., overweight, obesity), and body mass index (BMI); for cardio-metabolic aspects we considered metabolic syndrome (MetS), arterial hypertension, lipid profile, glucose levels, type 2 diabetes, insulin resistance, and uric acid. Included studies had to report at least one of these primary outcomes. 

### 2.3. Inclusion and Exclusion Criteria

For inclusion, studies were required to (i) include children and/or adolescents aged 2–18 years (or a mean within these ranges) as subjects of study. Studies that did not state the mean age of participants were classified as child or adolescent studies depending on the ages of the majority of the sample; (ii) have a defined measure of the child’s or adolescent’s breakfast consumption and/or breakfast skipping; (iii) be published in peer-reviewed journals in the English language; (vii) be published in the period January 2008–December 2018. We included studies even if they did not report completely the dietary quality data. We excluded intervention studies, studies where overweight/obese subjects were the only participants, and studies focusing on eating disorders.

### 2.4. Identification of Relevant Studies

Potentially relevant papers were selected by reading the titles and abstracts. If abstracts were not available or did not provide enough results the entire article was retrieved and screened to determine whether it met the inclusion criteria.

### 2.5. Data Extraction, Synthesis, and Quality Assessment

A form was generated to register whether individual studies met eligibility criteria and to collect data regarding the study design and methodological quality. Three investigators independently reviewed and extracted data from the papers according to the predetermined criteria. Any difference in opinion about the studies was resolved by discussion between all the investigators. The following data were extracted: author, date of publication, study design, characteristics of the participants (sample size, age, gender, and country), measures of breakfast behaviors, breakfast behavior (i.e., breakfast consumption/skipping), assessment methodology and reliability and validity of dietary measures, definition of weight excess and assessment methodology. This information is summarized in Table 1. About secondary outcomes also the following data were extracted: definition of Mets, blood pressure, lipid profile, glucose, insulin, and uric acid levels, indexes of insulin resistance. Parameters could be as continuous or dichotomous variables. If reported, data on nutrient intake quality were included. This information is summarized in Table 2.

Study quality was independently assessed by three reviewers according to the Newcastle-Ottawa Scale for quality assessment of cohort studies and case-control studies [36]. The scales allocate stars, with a maximum of nine; the criteria were quality of selection (maximum, four stars), comparability (maximum, two stars), and exposure (maximum, three stars). High quality was assessed for more than eight stars.

## 3. Results

The literature search identified 239 potentially relevant articles. After reviewing the titles and abstracts and the full-length articles, 39 articles were selected for closer assessment and then included in our analysis [11,12,13,14,15,16,17,18,19,20,21,22,23,24,25,26,27,28,29,30,31,32,33,34,35,36,37,38,39,40,41,42,43,44,45,46,47,48,49]. The search flow-chart is represented in Figure 1. Agreement between reviewers on which studies to include was good: the K for the agreement was 85% after screening titles and abstracts and 100% after screening full-text articles. Overall, data from a total of 286,804 children and adolescents, living in 32 countries, were reported (Figure 2). 

Study quality was reported in Appendix A. The risk of bias was relatively high because (1) exclusion and inclusion criteria were not always clear; (2) some of the studies were not gender- and age-balanced; (3) data on subgroups were difficult to be extrapolated; (4) some studies did not clearly describe how the allocation was performed in cases and controls (in particular for the definition of skipping breakfast); (5) some of the studies could have a selective reporting bias (mis-reporting or under-reporting of breakfast habits due to methods used for the dietary assessment); (6) methods for the definition of skipping breakfast, and OW/OB were heterogeneous or not reported; (7) methods for the evaluation of nutrition assessment were heterogeneous; (8) confounding factors were lacking or not clearly reported in most of the studies. 

### 3.1. Association of Skipping Breakfast with Overweight/Obesity

Thirty-seven out of 39 articles were selected for closer assessment of weight and then included in our analysis [11,12,13,14,15,16,17,18,19,20,21,22,23,24,25,26,27,28,29,30,31,32,33,34,35,36,37,38,39,40,41,42,43,44,45,46,47]. They are summarized in Table 1. Of the 37 selected papers, 32 were cross-sectional studies and 5 were longitudinal studies reporting cross-sectional data [11,13,23,29,33]. Overall, data from a total of 285,626 children were reported. They came from 33 different countries (Figure 2). Children’s age showed a wide range of variability ranging from 44 months to 21.2 years. Only one study included adolescents older than 18 years, and, although it partly failed to respect all the inclusion criteria, we did not exclude it because it was conducted in a school population; the age range depended on the repetition of grades and the population older than 18 years was only the 5.5% [12]. One of the studies included a sample of 9–15 years old, enrolled in 1985, with a follow-up of about 20 years (2004–2006) at ages 26–36 years [11]. One study included only preschool-aged children [13], 27 studies included only school-aged children and adolescents [1,11,14,15,16,17,18,19,20,21,22,23,24,26,27,28,31,32,38,39,40,41,43,44,45,46,47], and 9 studies analyzed both preschool-aged and school-aged children and adolescents [25,29,30,33,34,35,36,37,42]. Most studies recorded data about breakfast skipping by food frequency questionnaires [11,14,16,18,20,22,27,28,29,31,32,33,38,39,40,43,44,45,46,47], some on a recall-based methodology or by food diaries [13,15,21,24,30,34,36,37,42,46], in others yes/no answers or unspecified methods were used [11,12,17,19,23,25,26,35,41,44]. Questionnaires were administered to the children/adolescents or to the parents in the case of youngsters. 

The subjects’ weight and height were measured in most cases, whereas they were reported only in one study [16]. To define overweight and obesity, BMI *z*-scores or age- and sex-specific cut-offs according to international criteria were used in most studies, whereas national-specific growth charts were referred to in 4 papers [25,29,39,40].

The reported prevalence of breakfast skippers showed extremely wide variability, ranging from 0.7% to 74.7% according to the definition of breakfast skipping used. Seven studies did not report the prevalence of breakfast skippers [19,25,26,27,33,35,47]. 

Almost all the included studies concluded that skipping breakfast is associated with an increased risk or prevalence of OW/OB, in some cases with limitations: Mustaq et al. reported that skipping breakfast was associated with overweight only among girls [24], similarly to what reported by Januszek-Trzciąkowska et al. [28] with respect to obesity. Only 6 studies reported no association between anthropometric measures and the habit of skipping breakfast [25] for 3–10 years aged children [29,36,41,42]. Overall, a positive association between skipping breakfast and OW/OB was found in 270,362 subjects (94.7%).

In most studies the crude association between body fatness measures and skipping breakfast was considered, whereas in some others adjustments were performed for potential confounding factors, i.e., sex, age, ethnicity, smoking, dieting, physical activity, and parental education [12]; gender, family situation (single-parent family), ethnic background, education level and smoking [16]; eating fast food, and physical activity and sedentary lifestyle [24].

### 3.2. Association of Skipping Breakfast with Metabolic and Nutritional Aspects

The literature search identified 34 potentially relevant articles. After reviewing the titles and abstracts and the full-length articles, 6 articles were selected for closer assessment and then included in our analysis [11,45,46,47,48,49]. They are summarized in Table 2.

Of the 6 selected papers, 5 were cross-sectional studies and 1 was a longitudinal study reporting cross-sectional data [11,45,46,47,48,49]. Overall, data from a total of 16,130 children were reported. They came from 6 different countries. Children’s age showed a wide range of variability ranging from 6 years to 18 years. The longitudinal study included a sample of 9–15 years old, enrolled in 1985, with a follow-up of about 20 years (2004–2006) at ages 26–36 years [11]. Most studies recorded data about breakfast skipping by food frequency questionnaires [11,45,46,47,49], some on a recall-based methodology or by food diaries [46], in others yes/no answers or not specified methods were used [11,48]. Questionnaires were administered to the children/adolescents or to the parents in the case of youngsters.

In all studies, anthropometric data, such as weight, height, BMI, waist circumference, and blood pressure were collected [11,45,46,47,48,49], while in one study the percentage of body fat was also reported [47]. Blood samples for the evaluation of glucose metabolism and lipid profile were collected [11,45,46,47,48,49].

Metabolic syndrome (MetS) was described in 5 studies [45,46,47,48,49]. In some of the studies, MetS was defined using NCEP-ATP III criteria modified by Cruz and Goran for the pediatric age [45,48] whereas in one study NCEP ATP III adult criteria [47]. Ho CY et al. used specific criteria from Cook [46], while Osawa H et al. those identified by national scientific societies and the Ministry of Health, Labour and Welfare in Japan [49].

One study included an indicator of dietary quality named “The Youth Healthy Eating Index for the United States of America (US-YHEI)” modified to YHEI-Taiwan [46]. YHEI-TW scores was calculated using the 24-hour dietary recall and FFQs, with higher scores indicating a better diet. The items included whole grains, vegetables, fruits, dairy, meat ratio, snack foods, sweetened beverages, multivitamins, fried foods outside the home, consumption of breakfast and, dinner with the family.

The reported prevalence of breakfast skippers had a huge variability, ranging from 5.4% to 29.0% according to the definition of breakfast skipping used. Three studies did not report the prevalence of breakfast skippers [47,48,49].

Only 3 of the included studies [45,46,47] investigated the correlation between breakfast consumption and blood pressure, with a significant negative association in 2 of them [45,46].

In 4 out of the 6 studies [11,45,46,47], the correlation between breakfast consumption and lipid profile was evaluated. Subjects who skipped breakfast had lower HDL-cholesterol levels [45,46], increased triglycerides [45], total and LDL-cholesterol [11,45].

Only 3 of the studies [11,46,47] investigated glucose metabolism. Two of them reported a higher insulin-resistance in who skipped breakfast [11,47].

Almost all the studies concluded that skipping breakfast is associated with an increased risk or prevalence of MetS [45,46,47,48,49].

Ho CY et al. reported that children who consumed breakfast daily had higher intakes of saturated fat, cholesterol, vitamins A, B1, B2, calcium, phosphorus, magnesium, and potassium and better dietary quality in comparison with those who consumed breakfast <4 times per week. Diversely, breakfast skippers had the highest proportion (25.5%) of under-reporting energy intake than the controls [46]. 

## 4. Discussion

This systematic review confirms with more recent data and strengthens the evidence summarized in 2010 [6] indicating that children and adolescents who skip breakfast are at higher risk to be or become OW/OB. Skipping breakfast seems also associated with metabolic syndrome presence, but data are still anecdotal.

The last decades are characterized by an increasing incidence of pediatric obesity, determined by the change of many lifestyle factors, including the diet. The determination of more risky dietary habits has an impact on public health to plan prevention tailored programs.

We analyzed the studies on children and adolescents published in the last ten years, not covered by the previous systematic review [6] with the purpose of analyzing trends and increasing knowledge. Unexpectedly, we failed to find in children and adolescents intervention studies or RCTs aiming to analyze the causative effect of skipping breakfast on OW/OB and related comorbidities.

Firstly, we observed a prevalence of breakfast skippers ranging 1.3–74.7%, according to different definitions used for the skipping breakfast. However, most of the studies reported that at least 10–30% (mean ± SD: 16.0 ± 16.2%) of children and adolescents did not ever eat breakfast [11,13,14,15,16,18,20,21,22,24,26,27,28,29,30,31,32,33,34,36,37,38,39,40,42,43,44,45,46,47]. The data are consistent because the studies covered 286,804 pediatric individuals living in Europe, the US, Australia, New Zealand, Asia, and Africa. They included also pre-school age individuals, although data on them are difficult to extrapolate [25,29,30,33,34,35,36,37,42]. On the other hand, an increasing trend in skipping breakfast from childhood to adolescence is seen [21,22,43,44], as well as higher values in girls than in boys [28,37,42]. The prevalence is almost like to results reported in 2010 [6]. This means that, although epidemiological data on pediatric dietary habits are substantial, health plans to educate at a dietary day composed of 4–5 meals have been inefficacious, in particular in adolescents, or not portrayed yet. Furthermore, no studies investigated why breakfast was skipped, whilst some hypotheses have been widely discussed in literature including a lack of time, not feeling hungry in the morning, and weight concerns [50].

Secondly, some critical issues on methods should be underlined. In fact, the definition of skipping breakfast varied among all the studies, although the previous meta-analysis had already suggested improving this methodological topic in future studies to reach more significant results [6] and, recently, the American Heart Association proposed definitions to improve and make generalizable the research on this topic [51]. Most of the studies used a dummy (yes/no) or ordinal categories, based on the number of days without having a breakfast during the week [11,12,14,15,16,17,19,22,23,24,25,26,27,29,30,31,32,33,34,35,36,37,38,39,41,42,43,44,48,49]. On the other hand, some others used qualitative and unspecific categories (usually, often etc.) [13,18,20,21,28,40,45,46,47] making a comparison among the studies difficult. We can speculate that the metabolic balance reflects likely daily circadian rhythms influenced by prevalent meal schedules [52], but rigorous methods are needed to ascertain this hypothesis. Furthermore, the dietary history was collected with validated semi-quantitative or qualitative methods only in some studies [11,13,14,15,16,18,20,21,22,24,27,28,29,30,31,32,33,34,35,36,37,38,39,40,42,43,44,45,46,47,49]. The nutritional assessment is always difficult with varying degree of reliability and needs carefully handling, even more in children when parents or caregivers should be engaged. The lack of accuracy may increase the already in itself high prevalence of mis-reporting and under-reporting biases, mainly in obese individuals [53].

Despite several methodological limitations, skipping breakfast in children and adolescents is associated with OW/OB in most of the studies [12,13,14,15,16,17,18,19,20,21,22,23,24,25,26,27,30,31,32,33,34,35,37,38,39,40,43,44]. Only 6 studies did not report an association or reported it only in specific categories of subjects [24,28,29,36,41,42]. Even if these studies are comparable to the others with respect to the administration of questionnaires (self-reported completed by children or parents, or face-to-face interviews) and are not restricted to a specific country, several hypotheses could be considered. Firstly, most of them used a definition of skipping breakfast more general or referred only to the day before the study [24,29,36,41]. Since Coulthard JD and colleagues used the UK chart published in 1995 to define OW or OB, the cut-offs different from WHO or recent IOTF growth charts could have significantly contributed [42]. An interesting point refers to the lack of association with OW/OB in infants compared to older children. Kupers LK et al. failed to observe a higher risk of OW/OB in children aged 2 or 5 years [29], and Kuriyan R et al. a higher waist circumference in those younger than 10 years, but the association became significant only in those older than 10 years [25]. Moreover, in 3 out of 6 studies without an association with the risk of OW/OB breakfast skippers are mainly adolescent girls older than 14 years [36,41,42]. Coulthard JD and colleagues observed that most of the breakfast skippers were adolescent girls and that a higher proportion of girls in the 11-18-year-old age group stated that they were currently dieting than boys [42]. Moreover, 77.6% of the cohort published by Zalewska M et al. was composed of girls of 18 years old beyond the dietary assessment with a non-standard questionnaire [41]. Similarly, 61% of breakfast skippers of the studies of Fayet-Moore F et al. were again in the 14–18-year-old group [36]. On the other hand, two studies reported a higher risk of OW/OB only in females who skipped breakfast, but these populations are younger (5–12 years) [24,28] than the latter [36,41,42]. These outwardly contrasting data could be secondary to the fact that the time of puberty is characterized by the starting peak of OW/OB incidence and females enter before males in this period. Even so, in all, these findings suggest that in studies focusing mainly on infants or adolescents more attention to subgroups and related age-specific diseases is needed. The weight-control behaviors and puberty should be carefully considered, and this group should be analyzed separately. It is known that chronotype varies with age and environmental factors [52,54,55,56]. We can hypothesize that desynchronization of circadian rhythms does not still occur in infants and it is not so hitting on weight balance in infants. Conversely, in very young children the reason of skipping breakfast could vary including breastfeeding during the night, loss of appetite in a failure to thrive due to several diseases (malabsorption, sleep apnea etc.).

On the other hand, none of these studies can explain why skipping breakfast is associated with OW/OB. Some authors suggest that breakfast skipping is affected by collinearity with confounding factors such as sleep duration and quality, circadian rhythms with more eating in the evening, length of night fasting, and lower physical activity levels [6,42,52,55,56]. Some of the selected studies corrected for confounders [12,16,24], but more tailored study designs are needed. This suggestion is strengthened if we observed that Fayet-Moore and colleagues reported an association in a study [34] and failed to confirm it in another one [36]. The authors used in both the occasions data from the National Nutrition and Physical Activity Survey, conducted in 2007 [34], and 2011–2012 [36], respectively. Because the period between the two surveys is relatively short, not captured confounders seem to be the more realistic hypothesis.

All the aspects discussed (growth charts, the definition of skipping breakfast, age-dependent responses, gender, and confounding factors) are critical issues for the planning of future intervention trials.

As already anticipated, our review is not able to clarify if the OW/OB phenotype is secondary to a higher energy intake during the following hours in children who skipped breakfast. The data in the literature on this topic are few and controversial [6,42,52,57]. However, in the paper we selected, it seems that children and adolescents eating breakfast have a better nutritional profile in terms of micro- and macronutrients [13,36,42,44,46] and is in line with data on adolescents consuming 5 meals including breakfast [58]. How this aspect impacts on metabolism is still a matter of investigation.

Lastly, we aimed to evaluate if children and adolescents who were skippers had different metabolic phenotype. Unexpectedly, the published papers are few, counting only 6 studies, but enclosed about 16,000 children. This is surprising if we consider the long-lasting impact of OW/OB on metabolic and cardiovascular diseases.

All the studies included reported that breakfast skippers had a worse lipid profile and/or a higher prevalence of MetS [11,45,46,47,48,49]. Two studies also reported higher insulin [11,47] and another higher blood pressure levels [46] than those that ate breakfast regularly. All these results mirror mournfully those in adults [51].

The findings on higher cholesterol, triglycerides, and blood pressure levels have been mainly linked to higher insulin resistance in the morning [11,51]. Regarding cholesterol, higher fasting insulin levels may trigger higher hydroxyl methyl glutaryl Co-A (HMG-CoA) reductase expression, resulting in increased circulating concentrations [11]. On the other hand, because the nutrition quality of breakfast skippers seems worse, the role of higher intake of saturated fats, simple carbohydrates and salt in the other meals is another plausible explanation.

The increasing risk of MetS is likely associated again to higher insulin-resistance in breakfast skippers, but also to increased fat oxidation and low-grade inflammation [59,60]. Moreover, it could be partly mediated by a higher BMI, as recently shown in a meta-analysis of prospective cohorts for the risk of type 2 diabetes in adults [61]. On the other hand, Mets is strictly associated with other unhealthy lifestyles such as poor physical activity. However, it has been demonstrated that skipping breakfast adversely modulates clock and clock-controlled gene expression resulting in increased postprandial glycemic response in both healthy individuals and individuals with diabetes and MetS [62]. Some clock genes are associated also with lipid metabolism and the development of MetS [51], making the picture more colored and intriguing.

Although findings from observational studies cannot establish causality, more prospective and intervention studies from childhood to adulthood with the assessment of clinical end-points, including cardiovascular and metabolic diseases could provide insight into the associations we reviewed. In adults, contrary to observational data, skipping breakfast has not been found to be causal for obesity, metabolic alterations or hungry perception under the conditions so far examined in RCTs [57,63,64]. RCTs are needed to test the association in children and adolescents and confirm our recommendations. Moreover, authors should pay attention to all the confounders suggested and skipping breakfast, as well as nutritional assessment, should be defined and evaluated by using definitions and tools with an international agreement, as suggested by the American Heart Association [51]. The better investigation of all these aspects could help in validating “skipping breakfast” as a marker of risk of OW/OB. The “chrono-nutrition” has been investigated within the context of obesity since studies have shown that eating at the “wrong” time of day can induce weight gain, despite a similar caloric intake [55]. In this context, the role of skipping meals, in particular, breakfast that stops the night fasting, caught the eyes of researchers. On the other hand, many other players have been still poorly investigated, and in the future could explain some of the controversial results. Among the others, also the nutrient content of the daily meals could have an impact on the desynchronization of circadian rhythms implicated in the obesity increase [52]. Moreover, food timing seems to have a higher heritability, also for breakfast [65]. The length of night fasting, the presence of time-restricted feeding, the composition of nutrients of the last meal before sleeping, the chronotype of individuals, the food timing behaviors of parents and sibling should be all investigated with respect to breakfast skipping or not in further studies to reduce residual confounders beyond sleeping habits, daily food quality and physical activity. Since high plasticity and reprogramming of most metabolic patterns in pediatric age has been described [66], filling this gap is an unmet need.

With these limitations, our study provides data supporting skipping breakfast as a potential “marker” of lifestyle behaviors (yet to be elucidated) in children and adolescents that promote OW/OB and metabolic diseases. Additional studies would be helpful to standardize the definition and assessment method for breakfast skipping in order to establish the best practice for using it as a tool for assessing obesity risk.

## Figures and Tables

**Figure 1 nutrients-11-00387-f001:**
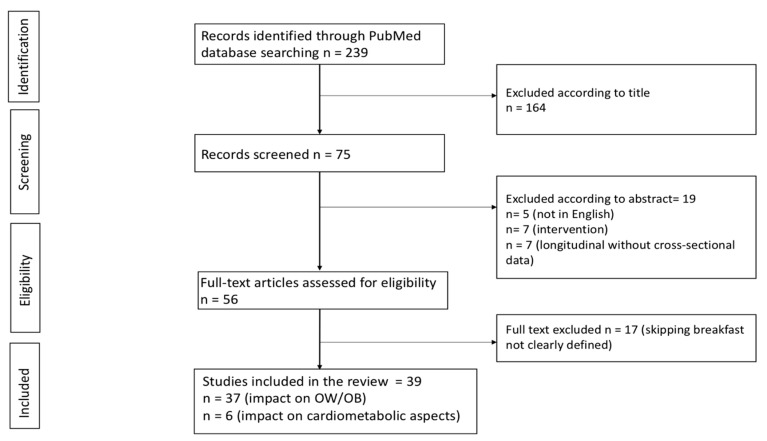
Flow diagram for study retrieval and selection.

**Figure 2 nutrients-11-00387-f002:**
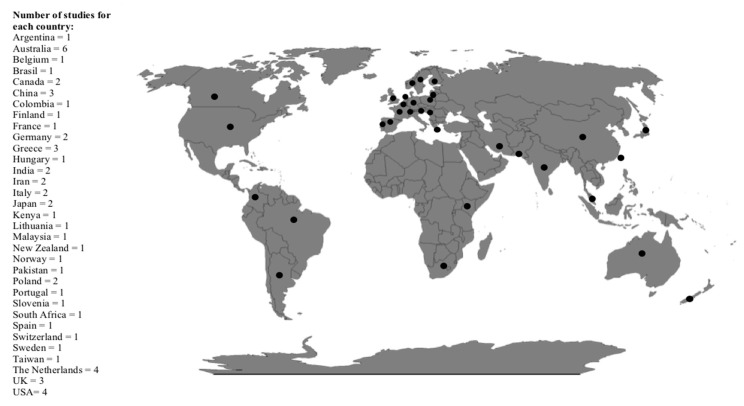
Countries’ distribution of the 39 selected studies.

**Table 1 nutrients-11-00387-t001:** Summarized studies’ characteristics on weight and skipping breakfast.

Reference (Author, year, n)	Study Subjects	Methods	Skipping Breakfast Definition	Breakfast Evaluation Method	OB/OW Definition	Prevalence of Breakfast Skippers	Results	Impact of Skipping Breakfast on OW/OB
Dialektakou 2008 [12]	N = 811, aged 14.9–21.2 years, M and F, Greece	Cross-sectional, self-reported questionnaires, measured height and weight	24 definitions evaluating breakfast consumption on the day of data collection, during the whole year, during the previous week, and on weekends/holidays	Not specified	Age- and sex-specific BMI cut-offs according to IOTF criteria	3.6–74.7% (according to different definitions)	Fewer breakfast-skipping variables associated with BMI than with OW/OB. Fewer associations when controlling for potential confounders. Fewer associations for variables corresponding to some definitions.	↑
Dubois 2008 [13]	N = 1549; aged 44–56 months, M and F, Canada	Longitudinal study, parent-report questionnaires, measured height and weight (cross-sectional data considered)	Frequency of breakfast eating: (1) yes, every morning; (2) regularly but not every day; (3) only on occasion; and (4) never. Categories 2 to 4 were classified as ‘breakfast skippers’	Eating behavior questionnaire (Enquete sociale et de sante’ aupres des enfants et des adolescents Quebecois -Health and Social Survey of Quebec Children and Adolescents) and a 24 h dietary recall interview	Age- and sex-specific BMI z-score cut-offs according to CDC criteria	10%	↑ intake of energy, carbohydrates or servings of grain products in breakfast skippers	↑
Harding 2008 [14]	N = 6599, aged 11–13 years, M and F, UK	Cross-sectional, self-reported questionnaires, measured height and weight	Number of eaten breakfasts per week (Every day; 3–4 days a week; 1–2 days a week; Never or hardly ever)	http://www.sphsu.mrc.ac.uk/studies/dash/ Food frequency questionnaire	Age- and sex-specific BMI cut-offs according to IOTF criteria	32.6–53% not eating breakfast every day	Increased risk for obesity in breakfast skippers (girls OR 1.74, 95% CI 1.30–2.34; boys OR 2.06; CI 1.57–2.70)	↑
Duncan 2008 [15]	N = 1229, aged 5–11 years, M and F, New Zeland	Cross-sectional, proxy questionnaire administered to the parents, measured height and weight, BIA	Number of eaten breakfasts in the last full week (0–7 days per week)	7-day recall	Boys and girls were classified as “overfat” if their % BF exceeded 25% and 30% (respectively)	1.3 (non-overfat)–2.6% (overfat) never eat breakfast in a week	Breakfast skippers had increased odds of overfat compared with those who had breakfast for five or more days/week.	↑
Croezen 2009 [16]	N = 25176, aged 13–16 years, M and F, The Netherlands	Cross-sectional, detailed Internet questionnaire, under supervision of instructed teachers following a standardized protocol, self-reported body weight and height	Number of eaten breakfasts per week (0–7 days per week)	Food frequency questionnaire	Age- and sex-specific BMI cut-offs according to Cole’s definition	29.3–39.2%	Skipping breakfast >2 times/week associated with OW (adjusted OR 1.68 (CI 1.43–1.97) in 13–14 year-aged, and 1.32 (CI 1.14–1.54) in 15–16 year-aged subjects); skipping breakfast every day associated with OB	↑
Nagel 2009 [17]	N = 1079, aged 6.2–9.2 years, M and F, Germany	Cross-sectional, self-reported questionnaires compiled by children and parents, measured weight, height, upper arm and waist circumference, skin fold thickness	Breakfast consumption before school (yes/no)	Not specified	Age- and sex-specific BMI cut-offs according to IOTF criteria	13.4%	Breakfast skippers had increased risk for OW (OR 1.73, 95%CI 1.13–2.64) and OB (OR 2.50, 95% CI 1.19–5.29)	↑
Sun 2009 [18]	N = 5753, aged 12–13 years, M and F, Japan	Cross-sectional, self-reported questionnaires, measured height and weight	Frequency of eating breakfast: daily, almost daily, sometimes, and rarely	Food frequency questionnaire	Age- and sex-specific BMI cut-offs according to IOTF criteria	1.1% of boys and 0.7% of girls ate breakfast rarely	Skipping breakfast (i.e., eating breakfast rarely) was associated with OW (in boys only after adjustment for age, parental OW and lifestyle variables)	↑
Maddah 2010 [19]	N = 6635, aged 6–11 years, M and F, Iran	Cross-sectional, self-reported questionnaire given to the parents, measured weight and height	Breakfast skipping (yes/no)	Not specified	Age- and sex-specific BMI cut-offs according to IOTF criteria	Not reported	Higher prevalence of OW/OB in breakfast skippers than in breakfast eaters (boys: 23.6% versus 16.9%, girls: 23.5% versus 17.1%)	↑
Isacco 2010 [20]	N = 278, aged 6–10 years, M and F, France	Cross-sectional, self-reported questionnaire compiled by the parents in the presence of their child, measured weight, height, WC and skin fold thickness	Frequency of eating breakfast: every day, sometimes, never	Food frequency questionnaire	Age- and sex-specific BMI z-score cut-offs according to CDC criteria	1.4% never ate breakfast	higher BMI *z*-score, skinfolds and WC in breakfast skippers	↑
Deshmukh-Taskar 2010 [21]	N = 9659, aged 9–18 years, M and F, USA	Cross-sectional, self-reported data on 24-h recall methodology over two days (assisted by parent/caregivers for children aged 6 to 11 years), measured weight, height and WC	Breakfast skippers: those who consumed no food or beverages, excluding water, at breakfast	24-h recall http://www.cdc.gov/nchs/data/nhanes/dr-1-5.pdf.	Age- and sex-specific BMI z-score cut-offs according to CDC criteria	20% of children, 31.5% of adolescents	Breakfast skippers had higher BMI *z*-scores and a higher waist circumference than ready-to-eat cereal and other breakfast consumers. Higher prevalence of obesity in breakfast skippers than ready-to-eat cereal consumers	↑
So 2011 [22]	N = 11570, aged 9–18 years, M and F, Hong Kong	Cross-sectional, self-reported questionnaires, measured height and weight, and BIA	Breakfast skippers (ate breakfast 0–2 times/week); semi-skippers (ate breakfast 3–4 times/week); non-skippers (ate breakfast 5–7 times/week)	Rapid Dietary behavior Assessment questionnaire (daily and weekly dietary behaviors, validated against the 24 h recall nutrient intake data in a smaller sample)	Age- and sex-specific BMI cut-offs according to IOTF criteria	8% of primary school students and 14% of secondary school students	Breakfast skippers had higher BMI, BMI *z*-scores and BF% than their counterparts	↑
Tin 2011 [23]	N = 113457, aged 9–10 years, M and F, Hong Kong	Longitudinal, 2-year follow-up, self-reported questionnaires, measured height and weight (cross-sectional data considered)	Breakfast skippers those who chose ‘no breakfast at all’	Not specified	Age- and sex-specific BMI cut-offs according to IOTF criteria	5.3% of boys, 5.2% of girls	Higher mean BMI in breakfast skippers both at baseline (β 0.77, 95% CI 0.67–0.87) and 2 years later (β 0.86, 95% CI 0.78–0.95)	↑
Mushtaq 2011 [24]	N = 1860, aged 5–12 years, M and F, Pakistan	Cross-sectional, questionnaires administered to the children by senior medical students, measured height and weight	Skipping breakfast once or more in the past week	7-day recall	BMI *z*-scores calculated based on the WHO criteria	8%	Breakfast skippers were significantly more likely to be overweight (15% versus 9%) and obese (13% versus 7%) than breakfast eaters (*p* = 0.002). Skipping breakfast was associated with overweight among girls (*p* < 0.001). Skipping breakfast as independent predictor of OW (OR 1.82, 95% CI 1.22–2.71)	↑ (OW in girls)
Kuriyan 2012 [25]	N = 8444 (4707 aged 3–10 years; N = 3737 aged 10–16 years), M and F, Bangalore	Cross-sectional, parent/student-report questionnaires, measured height and weight, WC	Breakfast skipping (yes/no)	Not specified	Indian Academy of Pediatrics cut-off for BMI; WC > 75th percentile for classifying abdominal obesity	Not reported	-	⇔ WC in children aged 3–10 years ↑ WC in children aged 10–16 years
Kyeariazis 2012 [26]	N = 2374, aged 6–12 years, M and F, Greece	Cross-sectional, self-reported questionnaires, measured height and weight	Breakfast skipping (yes/no)	Closed format questions in the form of multiple choice Questions	Age- and sex-specific BMI cut-offs according to Cole’s definition	Not reported	Skipping breakfast had a positive association with OB	↑
Van Lippevelde 2013 [27]	N = 6374, aged 10–12 years, M and F, Belgium, Greece, Hungary, the Netherlands, Norway, Slovenia, Spain, and Switzerland	Cross-sectional, self-reported questionnaires compiled by the children during school-time, measured weight and height	Breakfast frequency per week (0–7) calculated by adding up the breakfasts usually had on schooldays per week (0–5) and on weekend days per week (0–2)	http://projectenergy.eu Food frequency questionnaire	BMI *z*-scores calculated based on the WHO criteria	Not reported	Children’s breakfast consumption negatively related to children’s BMI-*z*-score	↑
Januszek-Trzciąkowska 2014 [28]	N = 2571, aged 7–9 years, M and F, Poland	Cross-sectional, self-reported questionnaire compiled by the parents, measured weight and height	Breakfast frequency: always, usually never	Food frequency questionnaire	Age- and sex-specific BMI cut-offs according to IOTF criteria	10.3% in girls, 9.1% in boys	Increased OB risk in girls irregularly or never eating breakfast (always versus usually, OR 2.71, 95% CI 1.33–5.51; always versus never OR 1.63, 95% CI 1.08–2.47)	↑ only for girls
Kupers 2014 [29]	T1: 2 years of age; N = 1488 T2: 5 years of age; N = 1366 M and F, The Netherlands	Longitudinal; parent-report questionnaires; measured height and weight (cross-sectional data considered)	Breakfast frequency per week (0–7), categorized as “eating breakfast daily” (7 times per week) or “not eating breakfast daily” (<7 times per week)	Food frequency questionnaire	Age- and sex-specific BMI cut-offs according Dutch reference growth charts	At T1, 3.0% of the children did not eat breakfast daily; at T2, 5.3%	No association between skipping breakfast and overweight, neither at age 2 nor at age 5	⇔
O’Neil 2015 [30]	N = 14200, aged 2–18 years, M and F, USA	Cross-sectional, self-reported questionnaires (complied by parents/guardians of 2–5 year children; by 6–11 year children assisted by an adult; older children provided their own recall), measured weight and height	24-h dietary recall: no breakfast or 11 possible breakfast patterns	24-h dietary recall interviews using an automated multiple-pass method http://www.cdc.gov/nchs/data/nhanes/nhanes_03_04/DIETARY_MEC.pdf.	Age- and sex-specific BMI cut-offs according to CDC criteria	18.7%	Mean BMI *z*-scores were lower among consumers of five breakfast patterns (grain/lower fat milk/sweets/fruit juice, pre-sweetened ready-to-eat cereal/whole milk, soft drinks/fruit juice/grain/potatoes, ready-to-eat cereal/whole milk, and cooked cereal/milk/fruit juice), compared to breakfast skippers.	↑
Smetanina 2015 [31]	N = 3990, aged 7–17 years, M and F, Lithuania	Cross-sectional, self-reported questionnaires (parents of younger age (7–9 years old) participants filled-in the questionnaire at home and older children and adolescents filled-in it themselves at school), measured weight and height	breakfast eating frequency per week: “Everyday” (“Everyday” and “4–6 times per week”), “1–3 times per week”, and “Never”	Modified WHO questionnaires (conducted by Health behavior in School-aged Children (HBSC) and COSI study groups).Food frequency questionnaire	Age- and sex-specific BMI cut-offs according to IOTF criteria	Never eating breakfast: 6.2% in underweight, 6.5% in NW, 9.6% in OW/OB	The prevalence of subjects never having breakfast was significantly higher in OW/OB than in NW (9.6% versus 6.5%)	↑
Zakrzewski 2015 [32]	N = 6841, aged 9–11 years, M and F, Australia, Brasil, Canada, China, Colombia, Finland, India, Kenya, Portugal, South Africa, UK, US	Cross-sectional, self-reported questionnaires, measured height, weight and BF%	Breakfast frequency per week (separately for weekdays and weekend days). 1. Three-category definition: weekly breakfast frequency coded as rare (0–2 days per week), occasional (3–5 days per week) and frequent (6–7 days per week). 2. Two-category definition: weekly breakfast frequency recoded as less than daily (0–6 days per week) or daily (7 days per week).	Food frequency questionnaire	BMI *z*-scores calculated based on the WHO criteria	Breakfast consumption: 6.3% rarely, 27.7% less than daily	Frequent breakfast consumption was associated with lower BMI z-scores compared with occasional (*p* < 0.0001) and rare (*p* < 0.0001) consumption, as well as lower BF% compared with occasional (*p* < 0.0001) and rare (*p* < 0.0001).	↑
Wijtzes 2016 [33]	N = 5913, T1: 4 years of age T2: 6 years of age, M and F, The Netherlands	Longitudinal, parent-report questionnaires, measured height and weight, percent fat mass by dual-energy X-ray absorptiometry (at age 6 years) (cross-sectional data considered)	At age 4 years: weekly consumption of breakfast (“never,” “1–2 days per week,” “3–4 days per week,” “5–6 days per week,” and “every day”, coded as 1–5); At age 6 years: the number of days of breakfast consumption assessed separately for weekdays (coded as 0–5) and weekend days (coded as 0–2), and the scores were summed to calculate total weekly consumption (0–7). Breakfast skipping defined as consumption <7 days per week	Food frequency questionnaire	Age- and sex-specific BMI cut-offs according to IOTF criteria	Not reported	Breakfast skipping at age 4 years associated with increased % fat mass at age 6 years (β = 1.38; 95% CI: 0.36–2.40)	↑
Fayet-Moore 2016 [34]	N = 4487, aged 2–16 years, M and F, Australia	Cross-sectional, computer-assisted interview based on 24-h recall methodology over two days from participants or their caregivers, measured height and weight	Breakfast skippers were children who did not consume an energy containing food or beverage during breakfast on 2 recall days	24-h recall methodology	BMI *z*-score or centile adjusted for age and sex was calculated using the US CDC 2000 growth reference chart	4%	Higher prevalence for OW/OB in breakfast skippers than in breakfast consumers (21.2% and 23.2% versus 16.4% and 16.5%, respectively)	↑
Alsharairi 2016 [35]	T1 (2006): N = 4601, 2–3 age of years T2 (2008): N = 4381, 4–5 years of age, M and F, Australia	Cross-sectional and longitudinal study, face-to-face mother’s interview, measured height and weight	Breakfast consumption in the day of interview (yes/no)	Not specified	Age- and sex-specific BMI cut-offs according to IOTF criteria	Not reported	OB boys at T1 (OR 2.38, 95% CI: 1.04–5.43) and T2 (OR 2.32, 95% CI: 1.01–5.32) and OB girls at T2 (OR 2.26, 95% CI: 1.14–4.46) were more likely to skip breakfast compared with non-overweight	↑
Fayet-Moore 2017 [36]	N = 2812, aged 2–18 years, M and F, Australia	Cross-sectional, face-to-face interviews, measured height and weight	Breakfast skipping or eating during the 24 h prior to the interview day	24-h recall methodology	Age- and sex-specific BMI cut-offs according to the WHO criteria	9%	No associations between anthropometric measures and breakfast or breakfast cereal choice were found Breakfast skippers: ↑ higher saturated fat intake ↓ intakes of dietary fibers and most micronutrients (*p* < 0.001)	⇔
Smith 2017 [37]	N = 1592, aged 2–17 years, M and F, Australia	Cross-sectional, computer-assisted interview based on 24-h recall methodology (for 2–5 year children completed by an adult; for 6–8-years an adult was interviewed with help from the child; 9–11 year children were interviewed directly with assistance from an adult; 12–17-year were interviewed directly, with the adult remaining in the room for those aged 12–14 years); measured weight and height	Breakfast skippers if they did not define an eating occasion as ‘breakfast’ in the 24-h recall or the energy intake for the “breakfast” occasion was <210 kJ	24-h recall methodology	Age- and sex-specific BMI cut-offs according to Cole’s definition	11.8% of boys and 14.8% girls skipped on one day and 1.4% boys and 3.8% girls skipped on both days	The odds of skipping breakfast were progressively higher with increasing BMI category	↑
Gotthelf 2017 [38]	N = 2083, aged 9–13 years, M and F, Argentina	Cross-sectional, self-reported questionnaires compiled by children and parents, measured weight and height	Breakfast habit: eating breakfast on the day of the survey (yes/no). Frequency: always (6–7 days/week), sometimes (2–5 days/week), never (0–1 day/week).	Food frequency questionnaire	BMI *z*- scores calculated based on the WHO criteria	64.1% of students from peri-urban schools and 46.1% of students from urban schools	Among breakfast skippers, 40.7% of the girls and 54.7% of the boys were OW/OB. A higher probability of skipping breakfast was associated with obesity.	↑
Nilsen 2017 [39]	N = 2620, aged 7–9 years, M and F, Sweden	Cross-sectional, proxy questionnaire filled out by the parents or guardians, measured height and weight	Number of eaten breakfasts over a typical week (Every day; most days, i.e., 4–6 days a week; some days, i.e., 1–3 days a week; Never)	Food frequency questionnaire	Age- and sex-specific BMI cut-offs according to Swedish national growth reference	4.6%	Association between OW/OB and not having breakfast every day (OR 1.9 (CI 1.18–3.13))	↑
Kesztyus 2017 [40]	N = 1943, aged 7.1 ± 0.6 years, M and F, Germany	Cross-sectional, proxy questionnaire administered to the parents, measured height, weight and WC	4-point scale, the results were subsequently dichotomized for analyses (never, rarely versus often, always)	Food frequency questionnaire	Age- and sex-specific BMI cut-offs according to Swedish national growth reference; abdominal obesity as WHtR >0.5 or >0.47 for girls and 0.48 for boys	13.1%	Skipping breakfast associated with OW (crude OR 2.02 (CI 1.18–3.43)), OB (crude OR 1.94 (CI 1.03–3.66)), abdominal OB with WHtR >0.5 (crude OR 2.51 (CI 1.63–3.88)), abdominal OB with WHtR >0.47/0.48 (crude OR 2.20 (CI 1.58–3.07))	↑
Zalewska 2017 [41]	N = 1999, aged 18 years, M and F, Poland	Cross-sectional, self-reported questionnaires, measured height and weight	Breakfast habit: skipped, <8 AM, ≥8 AM	Not specified	BMI calculated based on the WHO criteria	25%	No difference in the prevalence of breakfast skippers between NW and OW/OB	⇔
Coulthard 2018 [42]	N = 1686, aged 4–18 years, M and F, UK	Cross-sectional, 4-day food diary to be completed by the children, or their parent for those aged 11 years and under, measured weight and height	Those consuming breakfast every diary day, those consuming breakfast on at least one but not all diary days, and those not consuming breakfast on any diary day	4-day food diary	Age- and sex-specific BMI cut-offs according to Cole’s definition (1990 UK charts)	19.9% of girls and 14.5% of boys	No differences in weight status by breakfast eating habits. The overall nutritional profile of the children in terms of fiber and micronutrient intake was superior in frequent breakfast consumers (micronutrients: folate, calcium, iron and iodine (*p* < 0.01)	⇔
Tee 2018 [43]	N = 8332, aged 6–17 years, M and F, Malaysia	Cross-sectional, self-administered questionnaire with assistance to children aged 10 years and above, proxy questionnaire administered to the parent for children aged 6 to 9 years; measured weight and height	Breakfast skippers (ate breakfast 0–2 days/week), irregular breakfast eaters (ate breakfast 3–4 days/week) and regular breakfast eaters (ate breakfast ≥5 days/week)	Food frequency questionnaire	BMI *z*-scores calculated based on the WHO criteria	9.3% in primary school children and 15.9% in secondary school children	Compared to regular breakfast eaters, the risk of being OW/OB was higher in 6–12 years boys who skipped breakfast (OR 1.71, 95%CI 1.26–2.32), in 6–12 years girls (OR 1.36, 95% CI = 1.02–1.81) and in 12–17 years girls (OR 1.38, 95% CI 1.01–1.90)	↑
Archero 2018 [44]	N = 669, aged 6–16 years, M and F, Italy	Cross-sectional, self-reported questionnaires compiled by the children during school-time, in the presence of a teachers and medical staff, measured weight and height	Breakfast skipping (yes/no)	Italian version KIDMED index, a questionnaire of dichotomous (positive/negative) items	Age- and sex-specific BMI cut-offs according to IOTF criteria	14.8% in primary school children and 21.9% in secondary school children	OW/OB skipped breakfast more frequently than NW (chi-squared 3.556, *p* < 0.04). Increased risk for OW/OB in non-Italian breakfast skippers (OR 16.05, CI 95% 1.93–133.27, *p* < 0.01)	↑
Smith 2010 [11]	T1 (1985): N = 6559; 9–15 years of age T2 (2004–2006): 26–36 years of age M and F, Australia	The Childhood Determinants of Adult Health (CDAH) study. T1: self-report questionnaires; were measured: height and weight. T2: self-report questionnaires; were measured: height, weight, waist WC and BP; a venous blood sample was collected for lipid profile and glucose metabolism	T1: Breakfast consumption was assessed by using the question “Do you usually eat something before school?” “Yes” or “no” T2: Skipping breakfast was defined as not eating between 06.00 and 09.00	T1: Not specified. T2: Food-frequency questionnaire	Age- and sex-specific BMI cut-offs according to Cole’s cut-off	Skipping breakfast: 14.2% in childhood; 27.5% in adulthood	In both childhood and adulthood: ↑ WC (mean difference: 4.63 cm; 95% CI: 1.72, 7.53 cm)	↑
Shafiee 2013 [45]	N = 5625, subjects aged 10–18 years; M and F, Iran	The third survey of the national school-based surveillance system (CASPIAN-III); parent-report questionnaires; were measured: height, weight, waist WC and BP; a venous blood sample was collected for lipid profile and glucose metabolism	Subjects were classified into three groups: “regular breakfast eater” (6–7 days/week), “often breakfast eater” (3–5 days/week), and “seldom breakfast eater” (0–2 days/week)	Likert scale questionnaire	Age- and sex-specific BMI cut-offs according to the WHO growth reference standards	The % of subjects classified as: “regular”47.3%, “often” 23.7% and “seldom”29.0%, breakfast eaters	↑ (*p* < 0.001)	↑
Ho 2015 [46]	N = 2401, elementary school children; M and F, Taiwan	Elementary School Children’s Nutrition and Health Survey in Taiwan (NAHSIT); self-report questionnaire; were measured: height, weight, WC and BP; a venous blood sample was collected for lipid profile and glucose metabolism	Breakfast consumption was assessed by using the question “How often do you eat breakfast in a week?” The answer could range from 0 to 7 times. The frequency was classified into three groups, including 0–4, 5–6, and 7 times per week	24-h recall; food-frequency questionnaire The Youth Healthy Eating Index for the United States of America (US—YHEI) modified to YHEI—Taiwan (YHEI—TW): indicator of dietary quality	Not reported	% Breakfast frequency (times/week): 5.4% (0–4) 5.9% (5–6) 88.7% (7)	↑ (Children who consumed breakfast daily: ↓ BMI (17.9 kg/m^2^; *p* = 0.009); ↓ WC (58.6 cm; *p* = 0.005))	↑
Marlatt, 2016 [47]	N = 367, subjects aged 11–18 years; M and F, Minneapolis	Cross-sectional study; self-report survey; were measured: height, weight, % body fat, and BP; a venous blood sample was collected for lipid profile and glucose metabolism	Breakfast consumption was expressed as average number of days/week that breakfast was consumed	Self-report survey using validated questions (Nelson MC, Lytle LA, 2009. Development and evaluation of a brief screener to estimate fast-food and beverage consumption among adolescents. J Am Diet Assoc; 109, 730–734; 24-h recalls	Age- and sex-specific BMI cut-offs according to the CDC Growth Charts, (2000)	Not reported	↑ BMI and % body fat	↑

Legend: % BF = Percentage Body Fat; BIA = Bioelectric impedance Analysis; BMI = Body Mass Index; BP = Blood Pressure; CI = Confidence Interval; CDC = Center for Disease Control and Prevention; COSI = Childhood Obesity Surveillance Initiative; IOTF = International Obesity Task Force; M = Males; F = Females; NW = Normal Weight; OB = Obesity; OR = Odd Ratio; OW = Overweight; WC = Waist Circumference; WhtR = Waist-to-Height Ratio; WHO = World Health Organisation; ↑ = Increased; ↓ = Reduced; ⇔ = Not Variation.

**Table 2 nutrients-11-00387-t002:** Summarized studies’ characteristics on metabolic variables and skipping breakfast.

Reference (Author, Year, n)	Study Subjects	Methods	Skipping Breakfast Definition	Breakfast Evaluation Method	OW/OB Definition	Prevalence of Breakfast Skippers	Association of Skipping Breakfast with OW/OB	Association of Skipping Breakfast with Blood Pressure	Association of Skipping Breakfast with Lipid Profile	Association of Skipping Breakfast with Glucose Metabolism	Association of Skipping Breakfast with Metabolic Syndrome	Association of Skipping Breakfast with Nutrient Intake
Smith2010 [11]	T1 (1985): N = 6559; 9–15 years of age. T2 (2004–2006): 26–36 years of age. M and F, Australia	The Childhood Determinants of Adult Health (CDAH) study. T1: self-report questionnaires; were measured: height and weight. T2: self-report questionnaires; were measured: height, WC and BP; a venous blood sample was collected for lipid profile and glucose metabolism	T1: Breakfast consumption was assessed by using the question “Do you usually eat something before school?” “Yes” or “no”. T2: Skipping breakfast was defined as not eating between 06.00 and 09.00	T1: Not specified T2: Food-frequency questionnaire	Age- and sex-specific BMI cut-offs according to Cole’s cut-off	Skipping breakfast: 14.2% in childhood; 27.5% in adulthood	In both childhood and adulthood: ↑ WC (mean difference: 4.63 cm; 95% CI: 1.72, 7.53 cm)	Not reported	↑ Total (mean difference: 0.40 mmol/L; 95% CI: 0.13, 0.68 mmol/L) and LDL-cholesterol (mean difference: 0.40 mmol/L; 95% CI: 0.16, 0.64 mmol/L)	In both childhood and adulthood: ↑ fasting insulin (mean difference: 2.02 mU/L; 95% CI: 0.75, 3.29 mU/L)	Not reported	Not reported
Monzani 2013 [48]	N = 489, subjects aged 6.7 to 13 years; M and F, Italy	Population-based, cross-sectional study; self-reported questionnaire; were measured: height, weight, WC, and BP; a venous blood sample was collected for lipid profile, uric acid and glucose metabolism	Breakfast consumption: yes/no	Not specified	MetS according to modified NCEP-ATP III criteria of Cruz and Goran	Not reported	Not reported	Not reported	Not reported	Not reported	In school-children aged 10.1–13 years: no breakfast consumption (OR = 5.0, 95% CI = 1.5–17.2, *p* = 0.02) was ↑ in those with MetS	Not reported
Shafiee 2013 [45]	N = 5625, subjects aged 10–18 years; M and F, Iran	The third survey of the national school-based surveillance system (CASPIAN-III); parent-report questionnaires; were measured: height, weight, waist circumference (WC) and blood pressure (BP); a venous blood sample was collected for lipid profile and glucose metabolism	Subjects were classified into three groups: “regular breakfast eater” (6–7days/week), “often breakfast eater” (3–5days/week), and “seldom breakfast eater” (0–2 days/week)	Likert scale questionnaire	Age- and sex-specific BMI cut-offs according to the WHO growth reference standards Metabolic syndrome (MetS) was defined based on the Adult Treatment Panel III (ATP III) criteria modified for the pediatric age group	The % of subjects classified as: “regular”47.3%, “often” 23.7% and “seldom”29.0%, breakfast eaters	↑ (*p* < 0.001)	↑ (*p* < 0.001)	↑ Triglycerides, LDL-cholesterol (*p* < 0.001) ↓ HDL-cholesterol	Not reported	↑ (OR 1.96, 95% CI 1.18–3.27)	Not reported
Ho 2015 [46]	N = 2401, elementary school children; M and F, Taiwan	Elementary School Children’s Nutrition and Health Survey in Taiwan (NAHSIT); self-report questionnaire; were measured: height, weight, circumference waist (WC) and blood pressure (BP); a venous blood sample was collected for lipid profile and glucose metabolism	Breakfast consumption was assessed by using the question “How often do you eat breakfast in a week?” The answer could range from 0 to 7 times. The frequency was classified into three groups, including 0–4, 5–6, and 7 times per week	24-h recall; food-frequency questionnaire. The Youth Healthy Eating Index for the United States of America (US-YHEI) modified to YHEI-Taiwan (YHEI-TW): indicator of dietary quality	MetS was defined based on criteria from Cook	% Breakfast frequency (times/week): 5.4% (0–4) 5.9% (5–6) 88.7% (7)	↑ (Children who skipped breakfast daily: BMI (17.9 kg/m^2^; *p* = 0.009); WC (58.6 cm; *p* = 0.005))	↑ (Children who consumed breakfast daily: systolic BP (97.0 mmHg; *p* = 0.007); diastolic BP (57.3 mmHg; *p* = 0.02) Children who consumed breakfast daily versus children who consumed breakfast 0–4 times per week: risks of high blood pressure (OR = 0.37, 95% CI = 0.19–0.71))	HDL-cholesterol (Children who consumed breakfast daily: ↑ HDL cholesterol (59.5 mg/dL; *p* = 0.03))	⇔	↑ (Children who consumed breakfast daily: prevalence of MetS (2.89%) Children who consumed breakfast daily versus children who consumed breakfast 0–4 times per week: risks of MetS (OR = 0.22, 95% CI = 0.09–0.51))	YHEI-TW scores (Children who consumed breakfast daily versus those who consumed breakfast 0–4 times per week: ↑ intakes of: saturated fat, cholesterol, vitamins A, B1, B2, calcium, phosphorus, magnesium, and potassium; ↑ YHEI-TW scores (better dietary quality))
Osawa 2015 [49]	N = 689, subjects aged 10–13 years; M and F, Japan	Cross-sectional study; self-report questionnaire; were measured: height, weight, WC and BP; a venous blood sample was collected for lipid profile and glucose metabolism	Breakfast consumption was assessed by using the question “Do you have breakfast every day? (Yes, alone/Yes, with family/Seldom/No)	Food-frequency questionnaire designed by members of the Ichikawa Dental Association	MetS was defined based on criteria identified by the Japanese Society of Internal Medicine, the Japan Society for the Study of Obesity and the Ministry of Health, Labour and Welfare in Japan	Not reported	Not reported	Not reported	Not reported	Not reported	Not eating breakfast was associated significantly with MetS or high risk MetS (OR: 2.70, 95% CI: 1.01–7.23, *p* < 0.05)	Not reported
Marlatt, 2016 [47]	N = 367, subjects aged 11–18 years; M and F, Minneapolis	Cross-sectional study; self-report survey; were measured: height, weight, BF%, and blood pressure BP; a venous blood sample was collected for lipid profile and glucose metabolism	Breakfast consumption was expressed as average number of days/week breakfast was consumed	Self-report survey using validated questions (Nelson MC, Lytle LA, 2009. Development and evaluation of a brief screener to estimate fast-food and beverage consumption among adolescents. J Am Diet Assoc; 109, 730–734; 24-h recalls	Age- and sex-specific BMI cut-offs according to the CDC Growth Charts, (2000) MetS was defined based on the Adult Treatment Panel III (ATP III) criteria	Not reported	↑ BMI and % body fat	⇔	⇔	↑ HOMA-IR	↑ MetS cluster score	Not reported

Legend: BMI = Body Mass Index; CI = Confidence Interval; F = Females; CDC = Center for Disease Control and Prevention; M = Males; MetS = Metabolic Syndrome; OB = Obesity; OR = Odd Ratio; OW = Overweight; ↑ = Increased; ↓ = Reduced; ⇔ = Not Variation.

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
