# Peer review of "A Systematic Review of the Association of Skipping Breakfast with Weight and Cardiometabolic Risk Factors in Children and Adolescents. What Should We Better Investigate in the Future?"

_nutrients, 2019, doi:10.3390/nu11020387_

Reviewer 1 Report

Thank you to the authors for submitting a paper on an important topic. Tremendous work was done to gather the information. Please see the suggestions and comments below:

Abstract:

1.    The sentence line 16 “Associations with cardio…” is not complete nor does it suggest anything about skipping breakfast – please edit accordingly

2.    Stating “pediatric worldwide population” is very strong: please soften this.

3.    Did you also include studies that only had dietary data? How did you define “poor dietary habits”?

Introduction:

1.    Line 36 The sentence starting with “The process of”… incorrect English and not politically correct

2.    Line 42 – change “as” to “a”

3.    Line 56- please delete “in this scenario”- not often used in scientific writing

Methods

-       In your inclusion criteria there is no mention that the papers needed to have dietary intake measures completed but yet you evaluated them on this outcome. Please explain.

Results

-       You included studies that were within the mean of 2-18 y but could have a range over 18 and younger than 2? Why is this?

-       What are “older subjects”?

-       Smoking was included as a confounder in children?- this is what others have reported I understand, I just find this surprising

-       How was diet quality assessed in Ho study? Perhaps ass this.

Discussion

I am not sure your paragraph about “chrono-nutrition” is necessary: although I understand your point, I think it should be removed from the paper.

Author Response

Reviewer 1

Thank you to the authors for submitting a paper on an important topic. Tremendous work was done to gather the information. Please see the suggestions and comments below:

1) Abstract. The sentence line 16 “Associations with cardio…” is not complete nor does it suggest anything about skipping breakfast – please edit accordingly.

Answer. Done.

2) Abstract. Stating “pediatric worldwide population” is very strong: please soften this.

Answer. Done. We have cut the word “worldwide”.

3) Abstract. Did you also include studies that only had dietary data? How did you define “poor dietary habits”?

Answer. Thank you for the comment. No, we did not. We included also studies without dietary data. In fact, in methods, we wrote: “If reported, data on nutrient intake quality were included”. Only 5 studies of those included reported completely dietary data, although many more used FFQs. With “poor dietary habits” we mean low scores when used or low content in fruits, vegetables, fibers or high contents in sugars and saturated fats. We have rewritten the sentence in the abstract as follows: “Five studies reported a lower quality dietary intake in breakfast skippers”.

4) Introduction. Line 36 The sentence starting with “The process of”… incorrect English and not politically correct.

Answer. Thank you for the comment. We modified the sentence that now starts as follows “Overweight (OW) and OB result from…”.

5) Introduction. Line 42 – change “as” to “a”.

Answer. Done.

6) Introduction. Line 56- please delete “in this scenario”- not often used in scientific writing.

Answer. Done. We have substituted it with “basing on the above…”.

7) Methods. In your inclusion criteria there is no mention that the papers needed to have dietary intake measures completed but yet you evaluated them on this outcome. Please explain.

Answer. Thank you for the comment. We have better explain this point now.

8) Results. You included studies that were within the mean of 2-18 y but could have a range over 18 and younger than 2? Why is this?

Answer. Thank you for the comments. We did not include children younger than 2 years. The youngest children are aged 2 years [see ref 30, 34, 36, 37]. One study reported age in months but the youngest had 44 months, indeed more than 2 years of age [ref 13]. On the other hand, we decided to include a study with some subjects older than 18 years. This decision derived by the fact the Authors studied a very large population and the percentage of those aged 18-21 years was little (5.5%). Moreover, the study included only schoolchildren: the fact that someone was older than 18 years depended by the repetition of some grades [ref 12]. We have explained that in results now.  

9) Results. What are “older subjects”?

Answer. Thank you for the comments. We have pointed out that they were “school-aged children and adolescents”.

10) Results. Smoking was included as a confounder in children? - this is what others have reported I understand, I just find this surprising.

Answer. References number 16 reported smoking as a confounder. The study was conducted in adolescents (13-16 years) and smoking habits could be frequent. Many of the selected studies included adolescents and this factor should be taken more into account in particular in the association with other unhealthy lifestyle behaviors.

11) Results. How was diet quality assessed in Ho study? Perhaps ass this.

Answer. Thank you for the comments. We have inserted a description of the score used by Ho study and the items of the score. Ho et al. used the 24-h dietary recall and a FFQ to calculate the score. We have inserted also this information.

12) Discussion. I am not sure your paragraph about “chrono-nutrition” is necessary: although I understand your point, I think it should be removed from the paper.

Answer. Thank you for the suggestion, however, we have finally decided not to cut it because it is useful to answer to some points discussed by reviewers 3 (skipping breakfast as a marker of the risk of OW/OB in a complex picture). We have rearranged it in this vision.

We have also observed that the Supplemental Table 1 was not attached in the file we received. We attached again it.

Reviewer 2 Report

Table 1 seems to do OK without carry over of headings to next page, but table 2 is difficult to read without headings at the top of each page.

Author Response

Reviewer 2.

1) Table 1 seems to do OK without carry over of headings to next page, but table 2 is difficult to read without headings at the top of each page.

Answer. Thank you for the comment. We asked the journal editor to insert the heading at the top of each page for both the tables in the publication layout in case of acceptance.

We have also observed that the Supplemental Table 1 was not attached in the file we received. We attached again it.

Reviewer 3 Report

The authors have done an admirable job of summarizing the findings of papers, since 2008, examining the association of breakfast skipping with overweight and obesity, diet quality and metabolic fitness (when reported) among children and adolescents.  The following comments/suggestions could improve the paper and increase its impact.

The title of the paper uses the language "effects of skipping breakfast" on ....  Given that the paper only summarizes observational studies, the title should reflect this and use language like "association of skipping breakfast"  ... 

Throughout the paper (including the tables) the authors sometimes say "effects, influence or impact" and in other places they use "association" to describe the relationship between breakfast skipping and variables of interest.  Given the observational nature of the data reviewed, the term "association" should be used throughout.

The concluding sentence is misleading as it implies that skipping breakfast is causal in promoting obesity and other metabolic disorders.  Given that a causal connection has not yet been established, the authors may wish to characterize breakfast skipping as a "marker" of lifestyle behaviors (yet to be elucidated) that promote obesity and metabolic disease. 

The authors report that there were 6 studies that did not show a significant association between breakfast skipping and overweight/obesity.  It seems that these studies should be carefully compared to the others which reported an association to examine what was different...the population? the method of assessing breakfast skipping? cultural practices? something else?  What were the differences, and were they consistent across the "no association" studies? Such an analysis might reveal clues about what role breakfast might play in the causal pathway to obesity, yielding potential factors to examine in a randomized trial.

Why did the authors not include intervention studies or RCTs?  And, if not included, why were they not discussed as part of the literature review?  Are there studies that have examined whether skipping breakfast causes disturbances in food intake regulation, energy balance and metabolic dysregulation?  This seems central to the issue of how to interpret the associations between skipping breakfast and risk of obesity, etc.

There is a growing literature reporting on RCTs examining the effects of skipping breakfast on energy balance and other variables in adults (e.g., see Am J Clin Nutr. 2014 Aug; 100(2): 507–513).  In adults, contrary to observational data reported in adults, skipping breakfast has not been found to be causal under the conditions so far examined in RCTs.  Although these studies were not in children or adolescents it still seems like they should be noted in the discussion.  They support the authors suggestion that controlled trials need to be done to establish causality.

Overall, it seems the paper supports the idea that breakfast skipping may be an easy to assess marker for obesity risk whether or not it is directly involved in causality.   If they presented breakfast skipping as a marker, they could suggest what additional studies would be helpful to standardize the definition and assessment method for breakfast skipping in order to establish a best practice for using it as a tool for assessing obesity risk. 

Author Response

Reviewer 3.

The authors have done an admirable job of summarizing the findings of papers, since 2008, examining the association of breakfast skipping with overweight and obesity, diet quality and metabolic fitness (when reported) among children and adolescents. The following comments/suggestions could improve the paper and increase its impact.

1) The title of the paper uses the language "effects of skipping breakfast" on ....  Given that the paper only summarizes observational studies, the title should reflect this and use language like "association of skipping breakfast"

Answer. Thank you for the comment. We have changed the title as follows: “A Systematic Review of the Association of Skipping Breakfast with Weight and Cardiometabolic Risk Factors in Children and Adolescents. What Should We Better Investigate in the Future?”.

2) Throughout the paper (including the tables) the authors sometimes say "effects, influence or impact" and in other places they use "association" to describe the relationship between breakfast skipping and variables of interest.  Given the observational nature of the data reviewed, the term "association" should be used throughout.

Answer. Done.

3) The concluding sentence is misleading as it implies that skipping breakfast is causal in promoting obesity and other metabolic disorders.  Given that a causal connection has not yet been established, the authors may wish to characterize breakfast skipping as a "marker" of lifestyle behaviors (yet to be elucidated) that promote obesity and metabolic disease.

Answer. Thank you for the suggestion. We have changed the last sentence as requested.

4) The authors report that there were 6 studies that did not show a significant association between breakfast skipping and overweight/obesity.  It seems that these studies should be carefully compared to the others which reported an association to examine what was different...the population? the method of assessing breakfast skipping? cultural practices? something else?  What were the differences, and were they consistent across the "no association" studies? Such an analysis might reveal clues about what role breakfast might play in the causal pathway to obesity, yielding potential factors to examine in a randomized trial.

Answer. Thank you for the comment. We have rewritten two paragraphs to better explain possible causes of the contrasting results, namely the methods to define “skipping breakfast”, the age range (infants, adolescent girls) in which different mechanisms could have a role (breastfeeding during the night, loss of appetite in a failure to thrive in infants vs behaviors to control weight in adolescent females), and uninvestigated confounders. Regarding the last point, we have discussed the contrasting data published by Fayet-Moore and colleagues on data in the Australian cohort referred to the 2007 and 2011-2012 surveys.

5) Why did the authors not include intervention studies or RCTs?  And, if not included, why were they not discussed as part of the literature review?  Are there studies that have examined whether skipping breakfast causes disturbances in food intake regulation, energy balance and metabolic dysregulation?  This seems central to the issue of how to interpret the associations between skipping breakfast and risk of obesity, etc.

Answer. We did not include intervention studies or RCTs because we found only two studies eligible in these categories. However, after the reading of the main text, they failed to cover the interest of this review, and as a consequence, we did not cite them.

The first one was the COMPASS study, a longitudinal non-randomized study. The intervention was the administration of school breakfast programs. In this case, OW/OB or metabolic assessments were not ascertained. Skipping breakfast was only reported associated with the desire of losing weight and/or lack of involvement in school sports, but if the population who desired losing weight is not described in any terms (Godin KM, et al. Examining Predictors of Breakfast Skipping and Breakfast Program Use Among Secondary School Students in the COMPASS Study. J Sch Health. 2018 Feb;88(2):150-158)

The second one was a pilot cross over trial with 2 types of breakfast or fasting in 28 children aged 9-13 years. The intervention was a spot intervention, not prolonged over-time. Moreover, they did not study weight or metabolism, but only feelings with respect to fatigue or the desire of eating something 2 h after slipping breakfast (Pereira MA et al, Breakfast Frequency and Quality May Affect Glycemia and Appetite in Adults and Children. J Nutr. 2011 Jan;141(1):163-8).

However, we have inserted a sentence on the lack of these type of studies. 

6) There is a growing literature reporting on RCTs examining the effects of skipping breakfast on energy balance and other variables in adults (e.g., see Am J Clin Nutr. 2014 Aug; 100(2): 507–513).  In adults, contrary to observational data reported in adults, skipping breakfast has not been found to be causal under the conditions so far examined in RCTs.  Although these studies were not in children or adolescents it still seems like they should be noted in the discussion.  They support the authors suggestion that controlled trials need to be done to establish causality.

Answer. Thank you for the comment. We have inserted this point at the end of the discussion.

7) Overall, it seems the paper supports the idea that breakfast skipping may be an easy to assess marker for obesity risk whether or not it is directly involved in causality.   If they presented breakfast skipping as a marker, they could suggest what additional studies would be helpful to standardize the definition and assessment method for breakfast skipping in order to establish a best practice for using it as a tool for assessing obesity risk.

Answer. Thank you for the suggestion. We have stressed this point in both the abstract and discussion.

We have also observed that the Supplemental Table 1 was not attached in the file we received. We attached again it.

Round  2

Reviewer 3 Report

The authors have done a nice job of responding to the initial review and their modifications have improved the manuscript.  There are still a number of edits that need to be made to remove awkward sentence structure and correct usage. 

E.g.:

Line 50, should be "has" not have.

Line 58, first four words are awkward.  Consider using 

Line 91, awkward...consider changing "We included studies although" to We included studies even if...

Line 141, add "it was" after bacause

Line 149, consider "unspecified" instead of not specified

Line 230, consider changing "in the purpose" to "for the purpose or with the purpose"

Line 271, consider changing "diversely by" to "compared to"

Line 274, add ..."older than 10 years" to end of sentence

Line 297, change "study's design" to "study designs"

Line 339, consider removing "reported in them"

Line 341, consider ending sentence at "RCTs" and creating new sentence making the point that RCTs are needed to test the association in children and adolescents.

Line 359, add the word "potential" before the word "marker"

Author Response

Answer to reviewers

A Systematic Review of the Association of Skipping Breakfast with Weight and Cardiometabolic Risk Factors in Children and Adolescents. What Should We Better Investigate in the Future?

Alice Monzani, Roberta Ricotti, Marina Caputo, Arianna Solito, Francesca Archero, Simonetta Bellone, and Flavia Prodam

Reviewer 3.

The authors have done a nice job of responding to the initial review and their modifications have improved the manuscript.  There are still a number of edits that need to be made to remove awkward sentence structure and correct usage.

1) Line 50, should be "has" not have

Answer. Done

2) Line 58, first four words are awkward.

Answer. We have deleted them.

3) Line 91, awkward...consider changing "We included studies although" to We included studies even if...

Answer. Done

4) Line 141, add "it was" after bacause

Answer. Done

5) Line 149, consider "unspecified" instead of not specified

Answer. Done

6) Line 230, consider changing "in the purpose" to "for the purpose or with the purpose"

Answer. Done

 7) Line 271, consider changing "diversely by" to "compared to"

Answer. Done

8) Line 274, add ..."older than 10 years" to end of sentence

Answer. Done

 9) Line 297, change "study's design" to "study designs"

Answer. Done

10) Line 339, consider removing "reported in them"

Answer. Done

 11) Line 341, consider ending sentence at "RCTs" and creating new sentence making the point that RCTs are needed to test the association in children and adolescents.

Answer. Done

12) Line 359, add the word "potential" before the word "marker"

Answer. Done

Thank you very much for all the comments. We have also inserted a new reference (65), regarding a meta-analysis published a few days ago.